# Interferon-β Suppresses Transcriptionally Active Parvovirus B19 Infection in Viral Cardiomyopathy: A Subgroup Analysis of the BICC-Trial

**DOI:** 10.3390/v14020444

**Published:** 2022-02-21

**Authors:** Heinz-Peter Schultheiss, Claus-Thomas Bock, Ganna Aleshcheva, Christian Baumeier, Wolfgang Poller, Felicitas Escher

**Affiliations:** 1Institute of Cardiac Diagnostics and Therapy, IKDT GmbH, 12203 Berlin, Germany; ganna.aleshcheva@ikdt.de (G.A.); christian.baumeier@ikdt.de (C.B.); felicitas.escher@charite.de (F.E.); 2Division of Viral Gastroenteritis and Hepatitis Pathogens and Enteroviruses, Department of Infectious Diseases, Robert Koch Institute, 13353 Berlin, Germany; bockc@rki.de; 3Institute of Tropical Medicine, University of Tuebingen, 72074 Tuebingen, Germany; 4Department of Cardiology, Campus Benjamin Franklin, Charité, Berlin-Brandenburg Center for Regenerative Therapies (BCRT), Universitaetsmedizin Berlin, Corporate Member of Freie Universitaet Berlin and Humboldt-Universitaet zu Berlin, 13353 Berlin, Germany; wolfgang.poller@charite.de; 5DZHK (German Centre for Cardiovascular Research), Partner Site, 10785 Berlin, Germany; 6Department of Internal Medicine and Cardiology, Campus Virchow-Klinikum, Charité, Universitaetsmedizin Berlin, Corporate Member of Freie Universitaet Berlin and Humboldt-Universitaet zu Berlin, 13353 Berlin, Germany

**Keywords:** human parvovirus B19, transcriptional activity, interferon beta-1b

## Abstract

Human parvovirus B19 (B19V) is the predominant virus currently detected in endomyocardial biopsies (EMBs). Recent findings indicate that, specifically, transcriptionally active B19V with detectable viral RNA is of prognostic relevance in inflammatory viral cardiomyopathy. We aimed to evaluate B19V replicative status (viral RNA) and beneficial effects in a sub-collective of the prospective randomized placebo-controlled phase II multi-center BICC-Trial (Betaferon In Chronic Viral Cardiomyopathy) after interferon beta-1b (IFN-β) treatment. EMBs of *n* = 64 patients with B19V mono-infected tissue were retrospectively analyzed. Viral RNA could be detected in *n* = 18/64 (28.1%) of B19V DNA positive samples (mean age 51.7 years, 12 male), of whom *n* = 13 had been treated with IFN-ß. Five patients had received placebo. PCR analysis confirmed in follow-up that EMBs significantly reduced viral RNA loads in *n* = 11/13 (84.6%) of IFN-ß treated patients (*p* = 0.001), independently from the IFN-ß dose, in contrast to the placebo group, where viral RNA load was not affected or even increased. Consequently, a significant improvement of left ventricular ejection fraction (LVEF) after treatment with IFN-ß was observed (LVEF mean baseline 51.6 ± 14.1% vs. follow-up 61.0 ± 17.5%, *p* = 0.03). In contrast, in the placebo group, worsening of LVEF was evaluated in *n* = 4/5 (80.0%) of patients. We could show for the first-time the beneficial effects from treatment with IFN-ß, suppressing B19V viral RNA and improving the hemodynamic course. Our results need further verification in a larger prospective randomized controlled trial.

## 1. Introduction

Viral infections of the heart represent a major cause of heart failure with potential for transition to the clinical picture of dilated cardiomyopathy (DCM). Human parvovirus B19 (B19V) is the predominant virus currently detected in endomyocardial biopsies (EMBs) of patients with inflammatory dilated cardiomyopathy (DCMi) [1,2,3]. 

B19V is a member of the genus Erythroparvovirus of the family Parvoviridae, harboring a linear, single-stranded DNA-genome that encodes for the non-structural protein NS1, two structural capsid proteins VP1 and VP2, and the small accessory 11 kDa and 7.5 kDa proteins of largely unknown function. To date, therapeutic options against B19V infection have not been established. Previous studies suggest that B19V presence in EMB is an unspecific bystander [4,5,6]. However, this conclusion does not address the entire issue, as inclusion criteria of these studies were solely based on B19V DNA loads. From today’s virological point of view, it is important to consider B19V detection in the heart in a more differentiated way.

Assessing the replicative status of the virus is the most accurate way to determine transcriptional activity of B19V in the myocardium [7,8]. Whereas latent B19V infection indicated by only viral DNA (viral genome) detection in EMBs has presumably no effect on the clinical course, it has been shown that transcriptionally active B19V characterized by viral RNA detection leads to an altered cardiac gene expression and is associated with progressive cardiac dysfunction and impaired survival [9,10]. 

In this study we aimed to investigate the B19V replicative status and beneficial effects in a sub-collective of B19V positive patients with chronic viral cardiomyopathy from the prospective randomized placebo-controlled BICC-Trial (Betaferon In Chronic Viral Cardiomyopathy) after interferon beta-1b (IFN-β-1b) treatment. 

## 2. Materials and Methods

A placebo-controlled phase II multicenter BICC-Trial (ClinicalTrials.gov number, NCT00185250), the first randomized trial on non-ischemic, chronic viral cardiomyopathy, was completed in 2005 [11]. In the BICC-Trial, in total *n* = 143 patients with symptoms of heart failure and EMB-based confirmation of EV, ADV, and/or B19V genomes in their myocardial tissue were randomly assigned to double-blind treatment [11]. Only patients with chronic viral cardiomyopathy and a history of heart failure >6 months (mean 39 ± 52 months) were included into the study. None of the patients had serum marker for acute B19V infection during the screening, baseline or follow-up period.

Patients received either placebo or antiviral treatment with IFN-β-1b for 6 months in addition to standard heart failure treatment. Data analysis was performed within the CRC Transregio 19 (NCT02970227, 1 January 2004) approved by the by Institutional Ethics Committee (Charité, Berlin, Germany). Informed written consent to participate and consent for publication was obtained from each study patient.

Patients were randomized into three groups: IFN-ß (4 × 10^6^ IU), IFN-ß (8 × 10^6^ IU), and placebo. Patients underwent EMBs before (baseline) and after IFN-ß treatment (follow-up, 12 weeks after treatment termination). Immediately after taking EMBs, samples were transferred to RNA later solution (Thermo Fisher Scientific, Waltham, MA, USA) stabilizing the nucleic acids of the EMBs [12]. DNA was extracted by Puregene Core Kit A (Qiagen, Hilden, Germany) according to manufacturer’s instructions [10,12,13]. Total RNA was isolated using TRIzol Reagent (Thermo Fisher Scientific, Waltham, MA, USA), treated with RQ1 RNase-free DNAse (Promega, Walldorf, Germany) to remove any traces of DNA, and reverse-transcribed to cDNA with High-Capacity cDNA Reverse Transcription Kit (Thermo Fisher Scientific, Waltham, MA, USA) using random hexamer primers according to the manufacturers protocol (Thermo Fisher Scientific, Waltham, MA, USA) [10,12,13]. Following cDNA synthesis or DNA extraction, samples were stored at −80 °C until further evaluation [14]. Nucleic acid concentration was measured by PCR-based Quantifiler Human DNA Quantification Kit (Thermo Fisher Scientific, Waltham, MA, USA) according to the manufacturer’s instructions [12,14]. 

Compared to placebo, in enterovirus (EV) and adenovirus (ADV) infections IFN-β-1b led to effective virus clearance in follow-up EMB after treatment, whereas B19V viral persistence was barely affected in B19V positive mono-infected patients. 

At this time, initial diagnosis for viral infections in these EMBs did not determine transcriptional activity of B19V (viral RNA). No distinction was made between latent and replicatively active (viral RNA) B19V, hypothesizing that this lack of evidence may have led to incorrect results. 

In the present study we evaluated the replicative status of B19V in EMBs in a sub-collective of the prospective BICC-Trial of B19V mono-infected patients. In samples that were still available, we aimed to analyze whether IFN-ß is effective in suppressing B19V replicative activity and improving clinical outcome. 

Nested polymerase chain reaction (nPCR) and quantitative real time PCR (qPCR) targeting the VP1/2 and NS1 regions of B19V were applied to detect B19V genomes (DNA) and VP1/2, NS1 mRNA. In brief, all PCR reactions were performed in a 96-well microtiter plate format ((Thermo Fisher Scientific, Waltham, MA, USA) using primers and TaqMan probes and PCR Universal TaqMan Master Mix (Thermo Fisher Scientific, Waltham, MA, USA) according to the manufacturer’s instructions on a QuantStudio 12 K Flex Real-Time PCR System (Thermo Fisher Scientific, Waltham, MA, USA). Following initial denaturation at 95 °C for 10 min, 40 cycles of thermal cycling at 95 °C for 15 s and 60 °C for 60 s were performed. B19V plasmids containing the VP1/2 and NS1 sequences at serial dilutions were included to standardize the system. Copy numbers of viral RNA were normalized by quantification of isolated total mRNA measured as expression of the house-keeping gene HPRT. Expression of HPRT was measured using predesigned primers and probe (TaqMan gene expression assay; Hs99999909_m1) (Thermo Fisher Scientific, Waltham, MA, USA) and, therefore, also served as an internal quality control for extraction efficiency or possible sample degradation. Serial dilutions (25–2 ng/µL) of reverse transcribed Total RNA Control (Human) (Thermo Fisher Scientific, Waltham, MA, USA) were used to quantify HPRT expression [7,15]. 

To compare continuous variables between two groups, nonparametric Wilcox-on-Mann-Whitney-Test or Kruskal-Wallis one-way ANOVA with Dunn’s post hoc test were used. A probability value of *p* < 0.05 was considered statistically significant. Statistical analysis and graph design was performed using GraphPad Prism 7.04 software (GraphPad Software Inc., La Jolla, CA, USA).

## 3. Results

### 3.1. EMB Analysis

EMBs of *n* = 64 patients with B19V mono-infected tissue were available in the present analysis. 

For distinction of B19V DNA and RNA transcripts, both fractions were extracted in a separate approach. Importantly, RNA fraction was applied to DNase treatment in order to prevent any contamination with DNA. As appropriate contamination control, 5.5 µL RNA were taken after DNase treatment and prior to cDNA synthesis and applied to PCR amplification. No PCR products were observed in all RNA samples after DNase treatment, giving evidence for the lack of DNA contamination.

Viral RNA could be detected by qRT-PCR in *n* = 18/64 (28.1%) of B19V DNA positive samples (mean age 51.7 years, 12 male), of whom *n* = 13 had been treated with IFN-ß (*n* = 4 in dosage 4 × 10^6^ IU, and *n* = 9 in dosage 8 × 10^6^ IU). Five patients had received placebo (Table 1).

As a result, qRT-PCR analysis confirmed in follow-up that EMBs showed significantly reduced viral RNA loads in *n* = 11/13 (84.6%) of IFN-ß treated patients (*p* = 0.001) (Figure 1). This was independently of the IFN-ß dose (dosage 8 × 10^6^ IU vs. 4 × 10^6^ IU *p* = 0.3). In contrast, in the placebo group, viral RNA did not change from baseline to follow-up in *n* = 2 patients and increased in *n* = 3 patients.

### 3.2. Hemodynamic Analysis

We analyzed the hemodynamic course of left ventricular ejection fraction (LVEF) performed by echocardiography with Simpson method at baseline and follow-up. In total, we could show a significant improvement of LVEF after treatment with IFN-ß (LVEF mean baseline 51.6 ± 14.1% vs. LVEF mean at follow-up 61.0 ± 17.5%, *p* = 0.03). In contrast, in the placebo group, worsening of LVEF was observed in *n* = 4/5 (80.0%) of patients (LVEF mean baseline 52.0 ± 20.0% vs. LVEF mean at follow-up 42.0 ± 17.9%) (Table 2).

In parallel, treated patients displayed statistically significant improvement levels of N terminal pro brain natriuretic peptide (NT-proBNP mean at baseline 548 ± 204 pg/mL vs. NT-proBNP at follow-up 319 ± 59 pg/mL, *p* = 0.001) and New York Heart Association (NYHA, *p* = 0.003) functional class in contrast to placebo (Table 2).

## 4. Discussion

In the present study a sub-collective of the prospective randomized BICC-Trial of B19V mono-infected patients was retrospectively analysed. Here, we could show for the first-time beneficial effects from treatment with IFN-ß in suppressing transcriptional active B19V and improve hemodynamic course 6 months after treatment. 

In the BICC-Trial we have shown that IFN-ß was safe in patients with myocardial enteroviral or adenoviral persistence and resulted in elimination of viral genomes and improved LVEF, whereas B19V viral persistence was barely affected in B19V positive mono-infected patients. Recent experimental and clinical data foster the important role of transcriptional active B19V [9,16]. 

A first non-randomized study showed that suppression of B19V transcriptional activity by nucleoside analogue treatment with telbivudine improved hemodynamic and clinical outcome significantly [17].

We and others have shown that B19V as a vasculo-tropic virus targets endothelial cells with consequent endothelial dysfunction. IFN-ß therapy initiated after vascular damage had occurred inhibited vascular leakage and concurrently decreased vascular permeability in endothelial cells, modulation of virally induced chronic endothelial damage-specifically, apoptosis and endothelial regeneration—showing a new mechanism for the anti-inflammatory action of IFN-ß [16,18]. Further underlying mechanisms of IFN-ß in cardiac B19V infections remain to be elucidated.

Our study has high and immediate clinical impact, since B19V is the most frequently found cardiotropic virus in EMBs. Whereas persistence of solely B19V DNA in EMB is without clinical relevance, transcriptional activity of B19V infection is known to be associated with impaired prognosis. 

## 5. Conclusions

We conclude that patients with transcriptional active B19V in EMB can profit from IFN-ß treatment. Our results underline the importance of molecular differentiation of myocardial B19V infections, including transcriptional activity, as a key approach for a specific anti-viral treatment. This is of particular importance because European Society of Cardiology recommendations support an anti-inflammatory treatment in DCMi when active cardiac infection is excluded [19]. Thus, for this clinical scenario, there is an unmet need to include the evaluation B19V viral RNA into routine EMB diagnostics. 

### Limitation of the Study

This was a sub-cohort from a prospective randomized study with a limited number of samples available, and as such a possible effect of selection bias cannot be denied. These results need further verification in a larger randomized controlled trial.

## Figures and Tables

**Figure 1 viruses-14-00444-f001:**
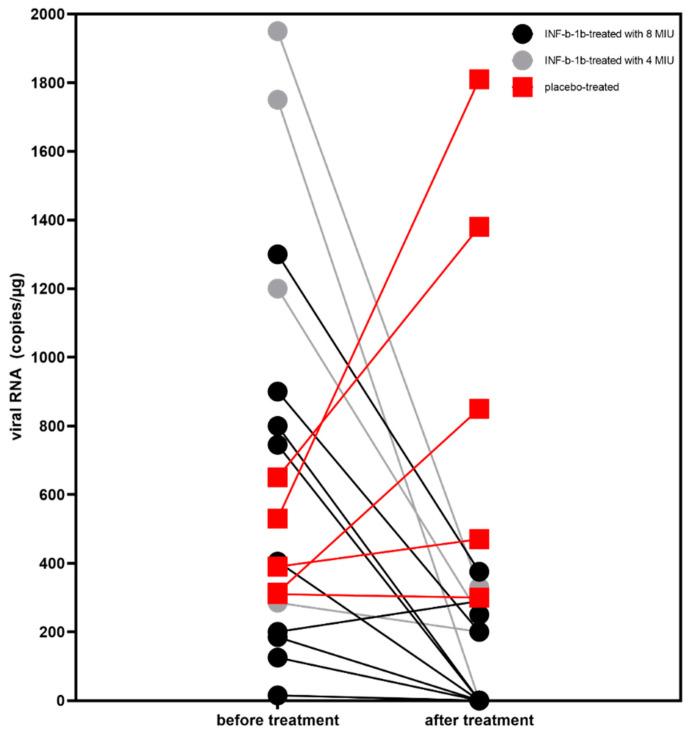
Changes in parvovirus B19 (B19V) viral RNA following 6 months IFN-ß treatment (determined as copies/μg). Patients received placebo are shown in red. B19V viral RNA was significantly reduced in IFN-ß treated patients from baseline to follow-up (*p* = 0.001). No significant difference was seen at baseline between INF-b treated and placebo patients (*p* = 0.3). In contrast, at follow-up IFN-ß treated and placebo patients differ significantly (*p* = 0.0005). Nonparametric Wilcoxon-Mann-Whitney-Test was used to compare the results at the two time points (before vs. after treatment) and between the groups. B19V: parvovirus B19; IFN-ß: interferon beta-1b; RNA: ribonucleic acid.

**Table 1 viruses-14-00444-t001:** Baseline Characteristics of Patients.

Variables	IFN-ß 4 × 10^6^ IU	IFN-ß 8 × 10^6^ IU	Placebo
Patient No., *n*	4	9	5
Sex, male	3	8	1
Age, years	61.00 ± 9.80	48.56 ± 13.07	50.00 ± 10.32
Duration of heart failure, month	26 ± 42	28 ± 44	31 ± 39
Systolic blood pressure, mm Hg	116 ± 11	120 ± 11	118 ± 6
Diastolic blood pressure, mm Hg	73 ± 5	73 ± 7	76 ± 7

Note: INF-ß: Interferon beta-1b; IU: International Units. The data are presented as mean ± standard deviation.

**Table 2 viruses-14-00444-t002:** Outcomes after 6-month treatment.

Variables	Treatment	Baseline	Follow-Up	*p*-Value (Baseline vs. Follow-Up)	*p*-Value (Treated vs. Placebo at Follow-Up)
LVEF, %	IFN-ß total	51.6 ± 14.1	61.0 ± 17.5	0.03	0.14
	Placebo	52.00 ± 20.0	42.00 ± 17.9	0.65	
NT-proBNP, pg/mL	IFN-ß total	548 ± 204	319 ± 59	0.001	0.004
	Placebo	514 ± 276	562 ± 202	0.53	
NYHA class (I/II/III/IV)	IFN-ß total	0/5/8/0	7/5/1/0	0.003	0.21
	Placebo	0/2/3/0	1/1/3/0	0.9	

Note: INF-ß: Interferon beta-1b total (4 × 10^6^ IU + 8 × 10^6^ IU). LVEF: left ventricular ejection fraction; NT-proBNP: N terminal pro brain natriuretic peptide; NYHA: New York Heart Association functional class. NYHA is a graded class of functional capacity. Class I indicates no symptoms or limitations in ordinary physical activity; class II, dyspnea with moderate exertion; class III, dyspnea with mild exertion; and class IV, dyspnea at rest. The data are presented as mean ± standard deviation.

## Data Availability

The data presented in this study are available on request from the corresponding author.

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
