# Peer review of "Interferon-β Suppresses Transcriptionally Active Parvovirus B19 Infection in Viral Cardiomyopathy: A Subgroup Analysis of the BICC-Trial"

_viruses, 2022, doi:10.3390/v14020444_

Round 1

Reviewer 1 Report

In this manuscript, Schultheiss HP et al. have analyzed B19V viral RNA expression in endomyocardial biopsies (EMBs) and interferon beta-1b treatment effects on patients presenting with this viral infection. The authors report that this treatment reduced B19 viral RNA expression in EMBs which was accompanied by an improvement in systolic function. On the other hand, patients with viral presence in EMBs in the placebo arm did not reduce viral RNA expression and exhibited worsened systolic function.

Main concerns:

-The method section needs to be expanded, including detailed information on the more relevant procedures performed in this study. Importantly, a paragraph explaining how the statistical analyses have been performed is mandatory.

-The results in Table 1 should include demographic and clinical variables associated with heart failure, for instance age, NYHA class, time since diagnosis of heart failure, NT-proBNP levels, renal function, etc… In addition, p-values should be specified (P<0.05, <0.01…?). Are there any differences in these variables between treated and placebo patients?

-In Figure 1, patients receiving different INF-beta doses should be represented differently (circles filled with different colors).

Reviewer 2 Report

Schultheiß and coworkers perform retrospective analysis of endomyocardial biopsies from a previously published (in multiple perspectives) placebo-controlled beta-interferon treatment trial. In the present manuscript their clinical cohort includes a small subgroup of biopsies concluded to be PCR-positive not only for B19-parvovirus (B19V) DNA but also for B19V-mRNA. During follow-up, the interferon treatment response (molecular; clinical) of such patients (n, 13) differed strikingly from the corresponding (placebo-treated) controls' (n, 5), whereby the authors conclude that transcriptionally active B19V infection constitutes an etiologically (and perhaps even therapeutically) important subset of cases of non-ischemic cardiomyopathy. This is a potentially significant finding; however, calling for major improvement of approaches and/or their description.

(1) The submitted text lacks information on the B19V-mRNA detection method used, as do the references cited to this end. After long search I did find ref. 7 (not cited in Methods) to contain such a description. Importantly, it turns out that the same primers were used for ex-vivo detection of both B19V-DNA and RNA. The RNA template was isolated by Trizol, followed by treatment with DNase (for DNA removal). After this, the RNA was reverse-transcribed to cDNA, for amplification with the said universal primers.

For distinction between DNA and RNA, a major question is the efficiency of the Trizol and DNase treatments. How do we know that the RT-PCR templates lacked any (residual) DNA? In our hands no DNase is sufficiently efficient to prevent all (any) PCR amplification. By which evidence were the heart-derived templates in the present study completely DNA-free after Trizol? Which DNase was used here? “DNase PeqLab, Erlangen, Germany” in Google fails to yield any essential information. Altogether, the authors should provide compelling evidence that their B19V-mRNA data were derived solely from viral mRNA and not from viral DNA.

(2) As of B19V, how was acute or recent infection excluded in the 18 patients? Were they viremic (B19V-DNA positive in blood, serum or plasma)? What was their in-depth B19V antibody status? Without experimental data the simplest explanation for the decreasing B19V-RNA loads during follow-up (Fig. 1) is convalescence after primary infection.

(3) The case definition (i.e. diagnostic terminology, both cardiological and virological) of these patients should appear in the manuscript more clearly. Is the diagnosis non-ischemic cardiomyopathy or dilative cardiomyopathy, and/or something else?

Round 2

Reviewer 1 Report

Taking into account the small number of patients, the authors should consider non-parametric tests for the analyses in Table 1. The authors should evaluate differences both at baseline and during follow-up.

¿How were differences in NYHA class compared ?

Reviewer 2 Report

The authors replied outstandingly to the questions/comments, whereby the revised manuscript is acceptable for publication.

For further (optional) improvement, the authors could additionally tell in their article, whether the key patients were (known to be) B19V-IgG-seropositive. (As they were B19V-IgM-negative).
